# ENHANCING VISION-LANGUAGE PROMPT LEARNING THROUGH IMAGE-TEXT DISTRIBUTION ALIGNMENT

## ABSTRACT

Large vision-language models (VLMs) such as CLIP have demonstrated impressive performance in zero-shot image classification tasks. These models usually leverage prompts to align the text and image distributions. However, existing prompting techniques have limitations in terms of interpretability or dynamic alignment of distributions. Specifically, the discrete prompt learning methods cannot effectively perform dynamic alignment of distributions, while the soft prompt learning method have very limited interpretability, rendering them challenging to comprehend and enhance. To jointly solve these issues, we leverage the interpretable descriptions to facilitate the soft prompt learning. In this paper, we introduce a novel training-free strategy to mitigate the distribution gap between plain text and image-text corpus, leveraging the power of pretrained models like GPT-3 to enhance image classification performance. Furthermore, we propose a new few-shot learning pipeline that incorporates a prompt learning and reweighting strategy to dynamically mitigate the image and text distribution gap. This method overcomes the limitations of existing prompting techniques and offers a more effective and interpretable solution for image classification tasks. Extensive experiments show the effectiveness of our method and illustrate the interpretability of our descriptions.

## 1 INTRODUCTION

Large Vision-language models (VLMs), such as CLIP (Radford et al., 2021), have shown remarkable ability on zero-shot image classification task. However, these VLMs necessitate prompts to effectively synchronize the text and image distributions. To this end, numerous methods for prompt learning have emerged (Zhou et al., 2022b;a; Menon & Vondrick, 2023), which aim to refine prompts to align with distinct image categories, thereby enhancing the accuracy of image classification. But those methods have their limitations. Discrete prompt learning methods struggle to dynamically align the image and text distribution for specific datasets, leading to sub-optimal results. In contrast, standard soft prompting methods, such as COOP (Zhou et al., 2022b), cannot provide an explanation of the reasoning process, as the prompts they learn tend to be meaningless. This deficiency undermines the reliability of predictions and impedes improvements of the results.

In our investigation, as depicted in Figure 1, a notable distribution discrepancy emerges in current prompt learning models and vision-language models. This misalignment is evident not just between the text corpus and the image-text corpus but also between the image-text corpus and the images themselves. This discrepancy holds significant implications for the broader realm of machine learning. Specifically, distribution misalignment can detrimentally impact the performance of VLMs, such as CLIP. When the text distribution internalized by these models does not align with the text corpus or certain image distributions, their effectiveness will be undermined. Our findings provide a new aspect that addressing this distribution gap has the potential to boost not only the performance of current models but also offers insights that can be universally applied to other prompt learning methods and models. The insights on distribution discrepancies derived from our work can serve as a foundational reference for enhancing future model designs and training paradigms.

In order to jointly tackle those challenges inherent in diverse prompting techniques, we propose a novel training-free strategy to minimize the shift in text distribution between plain text and image-text corpus. We first leverage the latent knowledge inherent in LLMs, such as GPT3 (Brown et al.,

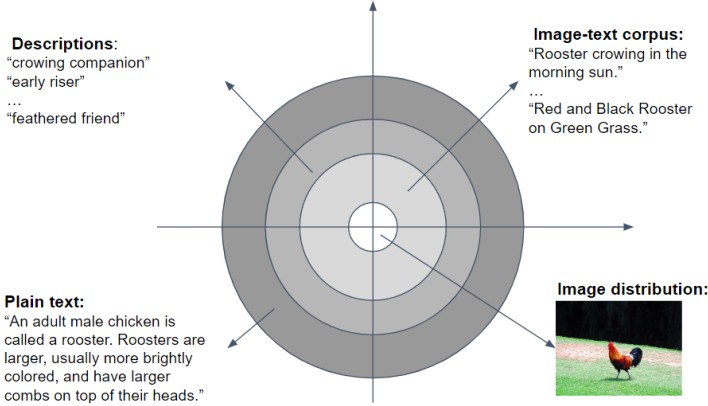

**Descriptions:**
"crowing companion"
"early riser"
...
"feathered friend"

**Plain text:**
"An adult male chicken is
called a rooster. Roosters are
larger, usually more brightly
colored, and have larger
combs on top of their heads."

**Image-text corpus:**
"Rooster crowing in the
morning sun."
...
"Red and Black Rooster
on Green Grass."

**Image distribution:**

Figure 1: An illustration of the distribution gap between images and different types of texts. The coordinate represents the embedding space. Each circle and ring represent their distributions. For example, the white circle indicates the image distribution, and the first outer ring represents the image-text corpus distribution. The closer to the origin, the similar distribution gap they have.

2020), to mitigate the distribution shift between the text corpus and the image-text corpus. These models have demonstrated remarkable capabilities in generating various texts, making them promising candidates for effectively mitigating the distribution gap between various of texts. Thus, we design a prompting method on GPT3 and GPT3.5, enabling them to generate semantic descriptions related to designated labels, while maintaining similarity to the image-text corpus that aligns with the distribution of the image-text model.

Furthermore, we present a few-shot reweighting strategy incorporating soft prompt learning methodology to further mitigate the distribution discrepancy between text and images. This approach adjusts the prompt weights based on their relevance to the target domain, ensuring a better alignment of the distributions and enhancing classification performance.

In summary, our contributions can be summarized as follows: (1) We design a zero-shot training-free strategy to generate descriptions that are interpretable and benefit for image-text retriever. (2) We propose a framework that learns soft prompts that dynamically align image distribution and text distribution while preserving the interpretability. (3) The extensive experiments show the effectiveness and interpretability of our method.

## 2 RELATED WORK

### 2.1 LARGE VISION-LANGUAGE MODELS

Large pre-trained models recently show great potential in representation learning, which have greatly advanced many downstream applications in natural language understanding and computer vision. Following the seminal work Transformer (Vaswani et al., 2017), many generative AI models emerge rapidly. The GPT series models (Radford et al., 2018; 2019; Brown et al., 2020; OpenAI, 2023; Touvron et al., 2023) have shown their powerful ability in text mining and reasoning. There are also many methods (Menon & Vondrick, 2023; Wei et al., 2021) that try to leverage the reasoning ability to improve downstream tasks. However, large language models are always trained on pure text data from Internet, so it is hard to align image distributions. On the other hand, large vision-language models (Radford et al., 2021; Li et al., 2022; 2023; Rombach et al., 2022; Jia et al., 2021; Singh et al., 2022) are able to bridge the images and texts in the latent space. Particularly, CLIP (Radford et al., 2021) is trained from a large set of image-text pairs, and it successfully mitigates the distribution discrepancy between text and images with the contrastive loss. It also shows tremendous zero-shot ability on the image classification task. However, its classification ability is highly dependent on the enormous training data. CLIP (Radford et al., 2021) takes the advantages of selected 400 million image-text pairs (Schuhmann et al., 2021), while ALIGN (Jia et al., 2021) takes 1.8 billion noisy image-text pairs. This training paradigm is largely dependent on training data. In other words, if the training data cover the specific domain, such as animals, food, or daily appliances, the model can perform well. However, when it comes to some unseen or uncommon domains, such as handwriting digits or medical images, the model would have quite poor performance. It reveals that even in large pre-trained models, it still has the domain gap issue. To address that issue, we come up with a method which leverages the reasoning ability of large language models, such as GPT-3 (Brown

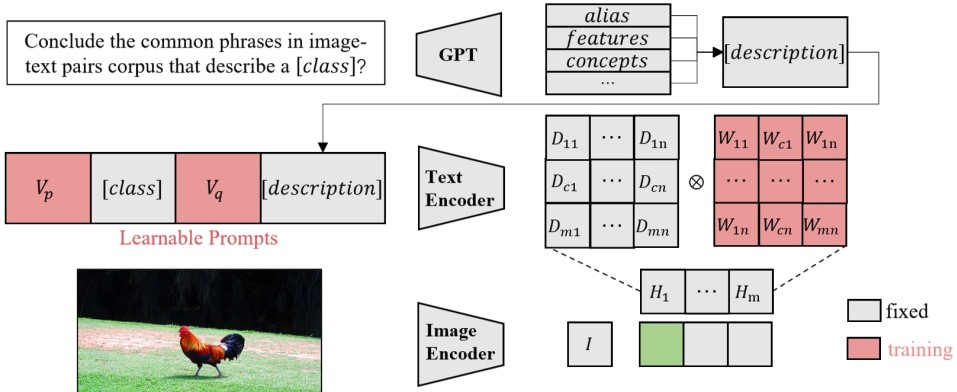

Figure 2: The overview of our proposed framework. Except for the learnable prompts $V_p$, $V_q$, and the learnable weights $W$, all other models, including GPT, Text Encoder, and Image Encoder, are fixed. The green color denotes the matched image and category. Initially, GPT is prompted to generate descriptions for each category. And then we learn soft prompts and learnable weights based on these descriptions to align the text distribution more closely with the image distribution, using the contrastive loss as a measure.

et al., 2020), to generate some descriptions that have semantic information, like features, alias, and some related concepts. In this way, it can mitigate the distribution shift between plain text category and image-text corpus, which is better aligned with the CLIP model distribution.

## 2.2 PROMPT LEARNING

Prompt learning is originated from the NLP area, which constructs prompts to help language models better understand questions. As the large language model is to predict the next words, the previous context plays a very important role in text generation.

While prompt learning achieves great performance on model tuning and downstream tasks, it is still underexplored in computer vision and visual language modeling. Pre-trained visual language models are incorporated with text information, and according to recent work (Radford et al., 2021; Zhou et al., 2022b), the text prompt also makes a difference on image classification. To exploit text prompt in classification problem, discrete prompts (Zang et al., 2022; Radford et al., 2021; Menon & Vondrick, 2023) are adopted to infer domain-specific knowledge. However, those methods may meet the sub-optimal issue which may not align with the specific domain distribution. To address this problem, some soft prompt learning methods (Zhou et al., 2022b;a; Hantao Yao, 2023; Lu et al., 2022; Zhu et al., 2022; Gao et al., 2021; Zhang et al., 2022) have been proposed. They take a few image samples to guide the text prompt to be aligned with the image distribution. For example, CLIP-adapter (Gao et al., 2021) adapts both image and text embeddings to a new embedding space. COOP (Zhou et al., 2022b) is the most related work to ours, which expands the labels with several learnable prompts and optimizes them with few-shot image samples. However, discrete prompt learning method may face the sub-optimal problem, while soft prompt learning methods have very limited interpretability. To overcome these issues, we propose a new method which jointly learns soft prompts while preserving the interpretability. We first generate some descriptions by a large language model which provides the prior knowledge and mitigates the pure text distribution and image-text distribution. Furthermore, we learn soft prompts to mitigate the distribution discrepancy between images distributions and text distributions. In this case, our method can preserve the prior knowledge of descriptions and provide explanations.

## 3 APPROACH

An overview of our framework is shown in Figure 2. In this section, we first briefly introduce our motivation and the overview of our framework. Then we present our technical details of our model in three main parts, namely Generating Descriptions; Prompt Tuning; Reweighting Aggregation.

### 3.1 OVERVIEW

Our objective is to enhance image classification using existinglarge pretrained model while simultaneously delivering interpretable explanations. To accomplish this, we introduce our framework, illustrated in Figure 2. This framework consists of two principal components. The first part is the static description generation, which leverages the reasoning ability of large language models, such as GPT-3 (Brown et al., 2020) to provide semantic description while aligning text distributions. The other part is the prompt learning pipeline. With the prior knowledge provided by large language model, few-shot images further guide the learnable prompts and learnable weights to adapt the description distribution to align with the image distribution.

### 3.2 GENERATING DESCRIPTIONS

Although the CLIP model shows its strong zero-shot ability, it still has domain bias due to the training data (Radford et al., 2021). Specifically, the distribution shift not only happens between images and texts, but also exists in natural languages. For example, a lot of natural language fine-grained categories, such as textures, landscapes, are less likely to appear in image-text corpus. The fact that the image-text corpus often comprises a higher frequency of keywords and a diminished occurrence of formal sentence structures reflects the distribution gap. (Schuhmann et al., 2021) In order to address the text distribution shift issue, we broaden the scope of labels by encompassing multiple descriptions which are more similar to image-text corpus and impart the prior knowledge by leveraging the reasoning and generative ability of large language models. Figure **??** illustrates the expanding details.

Given a particular category, our approach involves harnessing the capabilities of large language models to generate a set of text descriptions, denoted as $D$. These descriptions serve the purpose of enrich-

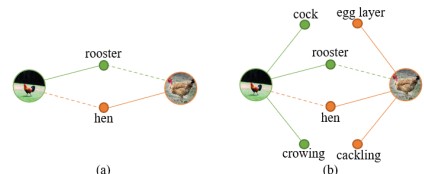

Figure 3: Illustration of how our descriptions work. (a) The original image-text retrieval solution is shown, where two images from different categories may have similar or even reversed similarity scores with their respective labels. For instance, an image of a rooster may have the same similarity score with both rooster and hen. (b) To address this issue, we expand the labels using their features, aliases, and concepts, leveraging prior knowledge and a closer image-text distribution. This approach leads to a higher error-tolerant rate, ultimately resulting in improved classification performance.

ing the label information while conforming to a distribution analogous to that of the image-text corpus. This process can be formulated as can be formulated as:

$$D_c = g(c), \tag{1}$$

where $g()$ represents any generic large language models, and $D_c$ refers to the descriptions regarding the category $c$. The descriptions will be employed to enhance the effectiveness of the visual language classification model.

Basically, we use LLMs to conclude several common phrases that are typically appeared in image-text corpus. Inspired by the prompt work (Menon & Vondrick, 2023), we design a text prompt to guide GPT-3 to generate related descriptions as:

*Q: Conclude the common phrases in image-text pairs corpus that describe a {category}.*
*A: Some of the most common phrases used to describe a {category} include:*

We also designed a similar prompt for ChatGPT. Further implementation details and comparisons can be found in Appendix.

Furthermore, it's essential to firmly anchor the generated descriptions to their corresponding labels. This is especially crucial in fine-grained classification tasks, where a single description might be applicable to multiple closely related categories. For example, "man's best friend" or " four legs" can describe all fine-grained dog categories, like wheaten terrier and scottish terrier. Thus, to narrow its meaning, we formulate a training-free zero-shot prompt as "an image of [CLASS], which relates to [DESCRIPTION]."

### 3.3 PROMPT TUNING

In our Prompt Tuning module, we aim to generate soft prompts for a set of images $X$ that can effectively bridge the domain gap between text and image distributions. Meanwhile, it should align with the latent distribution of CLIP. Consequently, as illustrated in Figure 2, we propose a description-based prompt tuning framework to reduce the distribution gap between the generated descriptions and images in a specific domain. Specifically, we introduce two soft prompts $V_m, V_n$ for the category label and its descriptions, respectively. Inspired by (Zhou et al., 2022b;a; Hantao Yao, 2023), we use the unified prompt for all classes and descriptions, which shares the soft prompts across all descriptions. The prompt is designed as:

$$t = [V_p][classname][V_q][description], \tag{2}$$

where $V_p$ is the class soft prompt and $V_q$ is the description prompt. Each $v \in \{V_p, V_q\}$ has the same dimension as the word embedding. $V_p$ and $V_q$ have $p$ and $q$ soft prompts respectively where $p$ and $q$ are hyperparamethers. Notably, each $t$ only corresponds to a single description.

Formally, we define the texture embedding encoded by CLIP text encoder $\theta$ as $D_{ci} = \theta(t_{ci})$, Where $c \in [0, m]$ refers to the class and $i \in [0, n]$ refers to the ith description corresponding to c. Additionally, m is the number of categories and n is the number of descriptions of one specific category. In this way, each prompted description will have one unique text embedding.

### 3.4 REWEIGHTING AGGREGATION

Existing work (Zhou et al., 2022b;a; Hantao Yao, 2023) primarily delves into prompting labels within a single text embedding. Nevertheless, our approach differs because we deal with scenarios where a single class could be described in multiple ways. This necessitates the aggregation of these descriptions into a single class embedding to compute similarity with image embeddings.

However, descriptions are typically generic and might not align perfectly with all images across different domains or datasets. Therefore, a simplistic averaging of all descriptions could still introduce significant domain shift. To address this, we further attenuate the distribution difference between text descriptions and image distributions, specifically focusing on a limited set of target domain images denoted as $X$.

We propose enhancing category representation by incorporating typical text description $d \in D$ that have a high correlation with the image-text corpus. This allows us to compute a more accurate and interpretable similarity score for a category via weighted summation.

In other words, we introduce a learnable weight matrix $W = R^{m*n}$ as shown in Figure 2. The weight matrix has the same dimension as the description embedding matrix. To aggregate the description embeddings into category embeddings, we perform a weighted summation as follows:

$$H_c = \sum_i^n D_{ci} \times \frac{W_{ci}}{\sum_{k=1}^n W_{ck}}, \tag{3}$$

where $D$ is the description embeddings and $W$ is the weight matrix. Once we obtain the category embeddings $H$, we are able to calculate the similarity of images and texts and reduce the contrastive loss:

$$L_{con} = -\sum_{x \in X} log \frac{exp(\phi(H_{c_x}, x))}{\sum_{i=1}^m (exp(\phi(H_{c_i}, x)))}, \tag{4}$$

where $c_x$ is the true category of the image $x$, $m$ is the number of categories, and $\phi()$ represent the similarity score of text embeddings and image embeddings. The similarity score of the correct category should be the higher than other categories in order to make right predictions.

## 4 EXPERIMENTS

In this section, we first introduce the experiment setting. Then we evaluate our approach in the following four problem settings: 1) evaluating the quality of descriptions. 2) training-free zero-shot image classification; 3) few-shot prompt tuning image classification; 4) evalutating the explainable results; and 5) Robustness evaluation.

| model | Average | | | DTD | | | Food101 | | | OxfordPets | | |
|---|---|---|---|---|---|---|---|---|---|---|---|---|
| | CLIP | CBD | Ours | CLIP | CBD | Ours | CLIP | CBD | Ours | CLIP | CBD | Ours |
| RN50 | 51.2 | 51.8 | **52.8** | 40.3 | 38.6 | extbf42.0 | 74.5 | 74.9 | **76.8** | 82.2 | 81.8 | **83.2** |
| VITB32 | 55.3 | 55.7 | **56.8** | 43.8 | 43.1 | **45.5** | 78.3 | 78.9 | **80.3** | 83.0 | 84.9 | **85.5** |
| VITB16 | 59.5 | 60.4 | **61.2** | 44.4 | 46.3 | **46.9** | 85.2 | 84.3 | **85.9** | 86.9 | 86.6 | **87.3** |

| model | Imagenet | | | Flower102 | | | FVGCAircraft | | | EuroSAT | | |
|---|---|---|---|---|---|---|---|---|---|---|---|---|
| | CLIP | CBD | Ours | CLIP | CBD | Ours | CLIP | CBD | Ours | CLIP | CBD | Ours |
| RN50 | 58.2 | 58.9 | **59.5** | 60.5 | 63.1 | **63.8** | 17.00 | 17.3 | **17.6** | 25.5 | **28.2** | 26.5 |
| VITB32 | 61.8 | 62.8 | **63.1** | 63.5 | 64.1 | **66.1** | 18.6 | **19.4** | 19.3 | **38.4** | 36.6 | 38.0 |
| VITB16 | 67.4 | 67.7 | **68.3** | 68.6 | 69.7 | **71.5** | 23.0 | 23.1 | **24.3** | 40.9 | **45.4** | 44.2 |

Table 1: Comparison of our training free model with two baselines. We test our model on 7 datasets with 3 CLIP backbones. Our results show consistent improvement across all settings. The number indicates the image classification accuracy. The best results are shown in **bold**.

## 4.1 DATASETS AND SETTINGS

**Datasets.** We employ 7 publicly available image classification datasets used in CLIP: Imagenet (Deng et al., 2009) OxfordPets (Parkhi et al., 2012), Flowers102 (Nilsback & Zisserman, 2008), Food101 (Bossard et al., 2014), FGVCAircraft (Maji et al., 2013), DTD (Cimpoi et al., 2014), and EuroSAT (Helber et al., 2019). These datasets constitute a comprehensive benchmark, which covers a diverse set of image distributions. For example, Food101, Flower102, and Oxford-Pets have images that are very common in daily life and dominant in CLIP training distributions. In addition, FGVCAircraft, DTD, and EuroSAT are less likely to appear in CLIP distributions and result in poor zero-shot performance. Imagenet, however, covers generic objects and fine-grained categories, which has a comprehensive neural distribution.

**Baselines** We compare our method with four existing baselines, including two zero-shot baselines and two prompt-tuning baselines. The zero-shot baselines are zero-shot CLIP (Radford et al., 2021), CBD (Menon & Vondrick, 2023). Zero-shot CLIP (Radford et al., 2021) is the original CLIP model. For a fair comparison, we adopt the standard handcraft prompt as "an image of a [CLASS]". CBD (Menon & Vondrick, 2023) is another training-free method that divides a category into several visual attributes. They further prompt these visual attributes and calculate the similarity score with images. Since our method can be done in the training-free setting, we compare our training-free version with these two training-free baselines. Additionally, the prompt tuning baselines are linear probe CLIP and COOP (Zhou et al., 2022b). The linear probe CLIP leverages the image embeddings generated by CLIP image encoder, and it trains a linear classifier on top of that. It also requires labeled training data. We followed the training process used by (Zhou et al., 2022b; Radford et al., 2021). COOP is the direct rival that learns soft prompts in a few-shot manner. It learns soft prompts to align image distributions.

**Implementation Details** For the description generation, we use GPT-3 (Brown et al., 2020) to generate descriptions. For the prompt tuning and reweighting aggregation method, we adopt 4/8/16-shot learning. We set $m, n$ as 4, so we have 8 shared learnable soft prompts. We initialize our learnable prompts by drawing from a zero-mean Gaussian distribution with standard deviation equal to 0.02. SGD is adopted as the optimizer, and an initial learning rate is set as 0.002, which is decayed by the cosine annealing rule. we use the warm-up trick by fixing the learning rate to 1e-5, as suggested in (Zhou et al., 2022b), only for the first epoch. We train 200 epochs for other datasets and 50 for Imagenet. To validate the generalization ability, we test our model on three CLIP backbones, RN50, VITB32, and VITB16. Our model is trained on an Nvidia A5000 GPU.

## 4.2 DESCRIPTION GENERATION

Figure 4 illustrates a selection of generated descriptions spanning both frequent, such as animals and less common domains, such as describable textures and fine-grained aircraft. On studying these descriptions, we discerned three primary types: features, aliases, and related concepts.

The feature type encapsulates both visual and non-visual attributes pertinent to a category. They provide supplementary information that can facilitate a more comprehensive understanding of categories for the CLIP model. For instance, descriptions such as "shattered lines", "jagged edges", "cracked lines", and "cracked surface" pertain to the "crack" texture, capturing nuances related to the

**samoyed**:
- snow-white pup,
- gentle giant,
- playful pooch,
- furry family member,
- cuddly cuddle buddy,
- smiling sammie,
- loyal companion,
- majestic majesty,
- happy-go-lucky pup

**rooster**:
- feathered friend,
- early riser,
- crowing companion,
- proud rooster,
- watchful guardian,
- feathered protector,
- barnyard beauty,
- cock-of-the-walk,
- barnyard buddy

**707-320**:
- passenger jet,
- long-range airliner,
- four-engine jet,
- iconic airliner,
- classic airliner,
- pioneering jet,
- pioneering aircraft,
- commercial airliner
- reliable workhorse

**cracked**:
- shattered lines,
- jagged edges,
- cracked lines,
- cracked surface,
- broken pieces,
- fractured fragments,
- splintered shards,
- shattered glass,
- broken shards

Figure 4: Examples of generated descriptions. We manually divided them into three types, namely features, alias, related concepts, which are shown in yellow, green, and black, respectively.

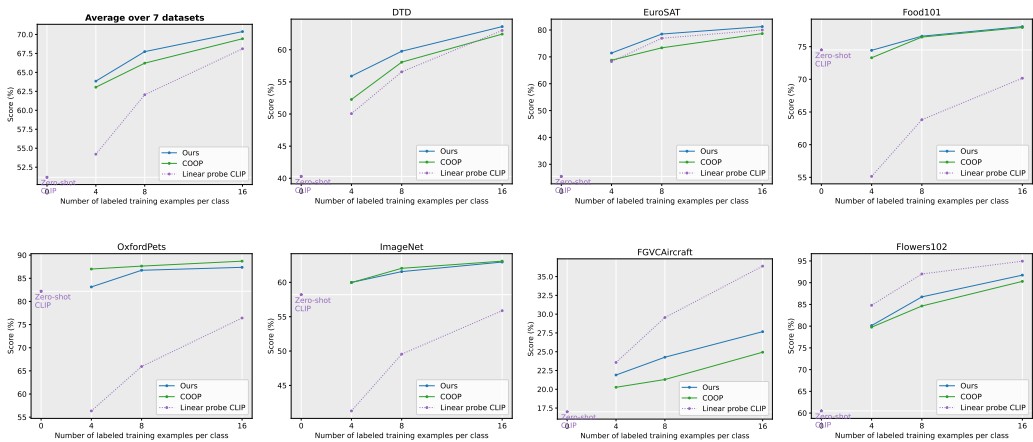

Figure 5: Main results of few-shot setting of our model on 7 datasets. We use RN50 as the CLIP backbone. Overall, our method can outperform other baselines. Comparing with the closest rival coop, we consistently outperform it over all datasets.

appearance of edges, the pattern of cracks, and the overall surface manifestation. This detailed feature aids the CLIP model in comprehending the composition and semantics of a category. The alias type incorporates alternate terminology likely to be employed in the image-text corpus. This broadens the label into multiple anchors, fostering a more comprehensive understanding. To illustrate, "707-320" refers to a specific aircraft model; however, devoid of context, it could be misconstrued as a number sequence or even a telephone number. By introducing aliases such as "iconic airliner" or "pioneering jet", we provide additional anchors, thereby reducing potential misinterpretations by the CLIP model. Lastly, the related concepts type includes common concepts that frequently occur in the image-text corpus. For instance, a "rooster" is often associated with a "barnyard". This association allows the model to leverage the additional context these concepts provide. In sum, our generated descriptions provide a higher level of interpretability and have closer distribution to image-text corpus, facilitating the CLIP model in better understanding and classifying images.

### 4.3 TRAINING-FREE ZERO-SHOT IMAGE CLASSIFICATION

In the training-free setting, the weights of each description are the same, so each description will equally contribute to the classification. For each category, it has 9 to 10 descriptions. For each description, we construct the zero-shot prompt template as "an image of [CLASS], which relates to [DESCRIPTION]." We compare our model with two baseline models, namely CLIP (Radford et al., 2021) and CBD (Menon & Vondrick, 2023), across three different architectures: RN50, VITB32, and VITB16. From the Table 1, we can observe that our training-free model consistently surpasses the performance of both baseline models across nearly all configurations and datasets. Specifically, our method showcases a consistent 3-5% improvement over the CLIP model, and also achieves 1-3% enhancement over CBD (Menon & Vondrick, 2023). Specifically, for certain datasets such as

| model | shots | DTD | | | EuroSAT | | | FVGCAircraft | | |
|---|---|---|---|---|---|---|---|---|---|---|
| | | CLIP | COOP | Ours | CLIP | COOP | Ours | CLIP | COOP | Ours |
| RN50 | 4 | 40.30 | 52.27 | **55.90** | 25.50 | 68.77 | **71.43** | 17.00 | 20.27 | **21.90** |
| | 8 | 40.30 | 58.07 | **59.77** | 25.50 | 73.37 | **78.50** | 17.00 | 21.30 | **24.27** |
| | 16 | 40.30 | 62.43 | **63.60** | 25.50 | 78.67 | **81.27** | 17.00 | 24.93 | **27.67** |
| VITB32 | 4 | 43.80 | 54.17 | **54.57** | 38.40 | **69.90** | 68.77 | 18.60 | 22.80 | **24.60** |
| | 8 | 43.80 | 59.87 | **60.97** | 38.40 | 74.40 | **75.50** | 18.60 | 23.90 | **27.30** |
| | 16 | 43.80 | 65.40 | **65.40** | 38.40 | 78.57 | **81.17** | 18.60 | 26.70 | **30.77** |
| VITB16 | 4 | 44.40 | 59.03 | **60.03** | 40.90 | **74.07** | 72.93 | 23.00 | 29.93 | **33.00** |
| | 8 | 44.40 | 63.90 | **64.33** | 40.90 | 77.97 | **78.47** | 23.00 | 32.20 | **36.73** |
| | 16 | 44.40 | 68.23 | **68.50** | 40.90 | 83.83 | **84.03** | 23.00 | 35.40 | **40.20** |

Table 2: Comparison of our prompt tuning model on few-shots settings with COOP (Zhou et al., 2022b). We test our model on 3 major uncommon datasets with 3 CLIP backbones to validate our robustness and effectiveness. Compared to baselines, our results show consistent improvement across all settings. The number indicates the image classification accuracy. Higher is better. The best results are shown in **bold**.

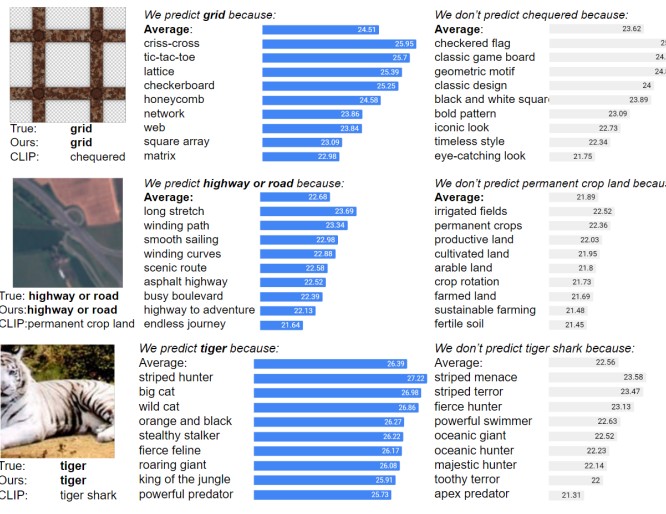

Figure 6: Examples predictions of our training free zero-shot model(left, blue) and CLIP model(right, grey). Visual comparison of predictions from both CLIP and our model with the ground truth labels beneath the respective images. The bar charts illustrate the descriptions corresponding to each category and the similarity scores of these descriptions. Our predictions demonstrate accuracy, and the similarity scores of descriptions possess the evidence to substantiate this claim detailing why our model avoids selecting incorrect labels.

DTD (Cimpoi et al., 2014) and flower (Nilsback & Zisserman, 2008), our method have around 6% improvement. This consistent improvent suggests that augmenting prior knowledge and bringing the image-text distribution closer can promote image-text alignment in the CLIP embedding space, leading to superior image classification accuracy. Additionally, our method exhibits noteworthy improvements in fine-grained and less common domains such as flowers and descriptive textures. In contrast, more prevalent domains like pets show relatively smaller enhancements. This suggests that our approach effectively applied across diverse image domains. For categories characterized by larger domain gaps, our method assists the CLIP model in comprehending semantic information by attenuating the domain shift. Conversely, for more popular categories, the benefits of our approach may be marginal, as these categories inherently align closely with the image distribution. These observations substantiate our hypothesis that the prediction accuracy of VLMs is significantly influenced by the domain gap.

## 4.4 FEW-SHOT PROMPT TUNING IMAGE CLASSIFICATION

In the few-shot domain prompt tuning setting, our objective is to further mitigate the distribution gap between images and descriptions, as there may exist some misalignment between them. To evaluate the effectiveness of our approach, we conducted experiments on seven different types of datasets, each representing distinct image distributions. The results, as shown in Figure 5, consistently confirm the superior performance of our model over COOP across all settings. Specifically, we have around 2% average improvement to COOP. Notably, the most signifiant improvement happens in FVGCAricraft (Maji et al., 2013), which has 11% improvement on average. This outcome

reinforces the notion that our techniques of description generation and reweighting indeed facilitate superior alignment between text and image distributions within the CLIP latent embedding space. Remarkably, our method exhibits even greater improvements on fine-grained and uncommon image datasets, such as FGVCAircraft, DTD, and EuroSAT, compared to popular image datasets. This observation highlights the robustness of our approach when dealing with significant distribution discrepancies. It implies that our model effectively leverages prior knowledge and aligns distributions more accurately, enabling enhanced performance on datasets with larger distribution gaps.

### 4.5 EXPLAINABLE QUALITATIVE RESULTS

Since we have incorporated additional descriptions, we can now interpret predictions based on the similarity between each description and the image. Figure 6 provides illustrative examples across different image types. We showcase some images from various domains including texture images (Cimpoi et al., 2014), satellite images (Helber et al., 2019), and animal images (Deng et al., 2009). These examples demonstrate how the additional information influences the final decision.

Consider an instance from the prevalent animal domain, where we encounter an image of a tiger. In the original CLIP model, the prediction might be incorrect due to the similarities between a tiger and a tiger shark at the textual level. However, by expanding the label with descriptions, CLIP gains a better understanding of the actual label. As we can see in the examples, descriptions of "striped hunter," "big cat," and "wild cat" exhibit relatively higher similarity to the image, correctly indicating the animal's classification as a tiger. Conversely, the descriptions associated with the tiger shark, such as "oceanic," result in lower similarity with the image, correctly distinguishing it as a different category.

On the other hand, as for the uncommon domains, our descriptions also provide some precise information. For example, the "criss-cross" show the features of the grid texture and meanwhile using the image-text manner to describe it. As a result, it obtains the highest similarity score and thus help the model making the right prediction.

These examples showcase how our approach enables CLIP to consider multiple pieces of evidence to make more accurate predictions. By leveraging descriptions and expanding the label space, CLIP gains a deeper insight into the visual content and context, allowing for improved classification performance across various image types and domains.

### 4.6 ROBUSTNESS EVALUATION

We examine the performance of our model on larger and more potent backbone CLIP architectures, such as VITB32 and VITB16. In order to validate the description ability and the robustness of our model on unseen domains, we select three less common image domains, which correspond to DTD (Cimpoi et al., 2014), EuroSAT (Helber et al., 2019), and FGVCAircraft (Maji et al., 2013). The results are presented in Table 2. Compared to the baseline CLIP model, our model demonstrates a considerable improvement in performance, validating our model's domain adaptation capability. The relatively poor performance of the CLIP model illustrates the substantial distribution discrepancy between images and texts. Our model, with the help of descriptions and learnable prompts, can bridge this distribution gap between images and texts, even in cases where the discrepancy is huge. Furthermore, our model surpasses the performance of the related rival model COOP (Zhou et al., 2022b), showcase the efficacy of our reweighting strategy. The consistent improvement across various domains also validates the robustness of our method in diverse and challenging environments.

## 5 CONCLUSION

In this paper, we aim to improve the prompting methods for visual language modeling and meanwhile enhance the interpretability. Specifically, we introduce a new framework for zero-shot classification with vision-language models in a domain alignment direction. Extensive results on multiple benchmark datasets show that our method provides explainable results, and it outperforms baselines on image classification. We further make use of the interpretable descriptions to align with image distribution by soft prompt tuning. It further improves the performance of our model while maintaining the interpretability.

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

**rooster**: (ChatGPT)
- rooster crowed,
- colorful bird,
- sunrise wake-up call,
- feathered friend,
- proud rooster,
- watchful guardian,
- farm fowl,
- barnyard buddy,
- strutting rooster,
- cock-a-doodle-doo

**rooster**: (GPT-3)
- feathered friend,
- early riser,
- crowing companion,
- proud rooster,
- watchful guardian,
- feathered protector,
- barnyard beauty,
- cock-of-the-walk,
- barnyard buddy

**cracked**: (ChatGPT)
- rough surface,
- chipped paint,
- weathered look,
- damaged texture,
- cracked pavement,
- peeling wallpaper,
- shattered ceramics,
- aged patina,
- broken glass,
- distressed appearance

**cracked**: (GPT-3)
- shattered lines,
- jagged edges,
- cracked lines,
- cracked surface,
- broken pieces,
- fractured fragments,
- splintered shards,
- shattered glass,
- broken shards

Figure 7: Qualitative comparison on descriptions generated by ChatGPT and GPT-3. Features, alias and related concepts are shown in yellow, green, and black, respectively.

# A   APPENDIX

## A.1   PROMPT GENERATION ON CHATGPT

Besides the GPT-3, we also evaluated the description quality on ChatGPT. Here is the prompt we provide.

*{"role": "user", "content": "Conclude the common phrases in image-text pairs corpus that describe a dog."},*
*{"role": "assistant", "content": """- man's best friend - furry friend - puppy love - good boy/girl - loyal companion - playful pup - fierce protector - canine cutie - furry family member - tail-wagging friend """},*
*{"role": "user", "content": "Conclude the common phrases in image-text pairs corpus that describe a cat."},*
*{"role": "assistant", "content": """- feline friend - purr-fect pal - cuddly companion - furry family member - playful kitty - curious critter - independent spirit - cattitude - four-legged friend - meow-velous mouser """},*
*{"role": "user", "content": "Conclude the common phrases in image-text pairs corpus that describe a [class]."}*

As illustrated in Fig. 7, analysis of the descriptions generated by the GPT-3 and ChatGPT models, it is evident that both models demonstrate proficient understanding and generation of contextually appropriate responses. They both successfully generate descriptions that capture typical characteristics and behaviors associated with the given keywords. Their ability to generate descriptions illuminates their capacity for enriching the understanding of the context. Since the descriptions generated by ChatGPT may not have consistent format in practice, we use descriptions generated by GPT-3 instead.

## A.2   BIDIRECTIONAL DOMAIN ALIGNMENT

While adapting text to image distribution have demonstrated substantial potential, they typically harness only a text modality. We assume that adapting the image modality to the target distribution could also result in superior distribution alignment. In pursuit of this, we explore the image variation model (Rombach et al., 2022). We employ a stable diffusion image variation model to generate similar images given an original image.

We synthesize a set of images similar to those in the distribution data, with the aspiration that this could serve as an augmentation method effectively representing image distributions. However, the quality of these images tends to be inconsistent. As depicted in Fig.8, for less common images like EuroSAT(Helber et al., 2019), by training the soft prompt using both these augmented images and the original one-shot image, we observe some performance improvement. However, when dealing with more common images, such as those in the OxfordPets (Parkhi et al., 2012) dataset, the gen-

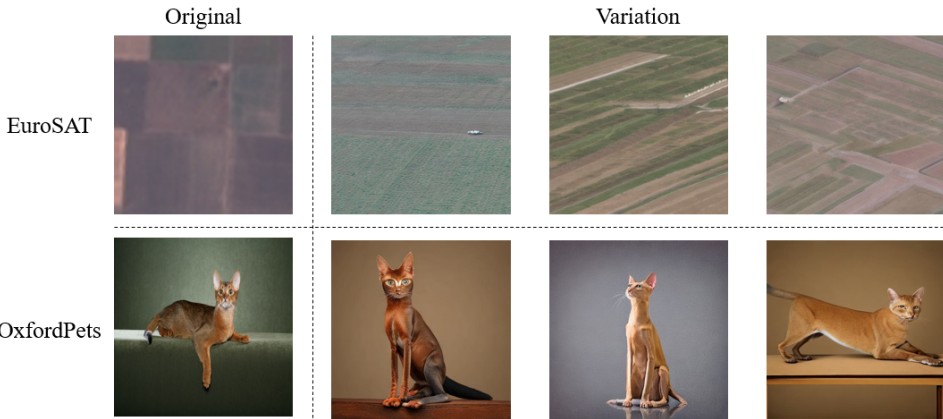

Figure 8: Examples of image variation quality. The left part is the in distribution data in original datasets and the right images are the generated images by image variation model given the single one-shot original image.

Table 3: Results of image classification accuracy based on augmented images. We tested the 1,2,4-shot image results.

| Dataset | 1-shot | 2-shots | 4-shots |
|---------|--------|---------|---------|
| EuroSAT | 53.47 | 53.97 | 57.97 |
| OxfordPets | 76.13 | 67.83 | 65.77 |

erated images fail to represent the in-distribution data, leading to a decrease in image classification performance. The detailed results of this investigation are presented in Tab. 3.

### A.3 LIMIATATIONS ON LLMS

While Language Learning Models (LLMs) exhibit impressive capability in rendering semantic information, there are nonetheless certain limitations that influence their outcome, particularly in the context of fine-grained image classification. For example, when encountering classes that demand intricate descriptions, LLMs occasionally fall short in generating meaningful descriptions. Instead, they tend to rendering generic descriptions that offer minimal distinctive information for fine-grained categorization. For instance, in OxfordPets dataset (Parkhi et al., 2012), around 20% categories have the description saying "man's best friend" which provide almost none additional information. One possible solution is providing additional contextual information specific to the domain under consideration. By doing so, LLMs could potentially filter out redundant descriptions, thereby enhancing their ability to generate more detailed and distinguishing descriptors. The exploration of methods to improve LLMs' performance in these more complex scenarios represents an interesting direction for future research.

