# OpenReview forum: "Enhancing Vision-Language Prompt Learning through Image-Text Distribution Alignment"
_ICLR.cc/2024/Conference — ICLR 2024 Conference Withdrawn Submission_

### Official Review · Reviewer_1JaY · 2023-10-20

**Soundness:** 2 fair
**Presentation:** 2 fair
**Contribution:** 2 fair
**Rating:** 3
**Confidence:** 4

**Summary:**

This paper addresses the limitations of existing prompting techniques used in large vision-language models like CLIP for zero-shot image classification. The authors introduce a novel approach that leverages descriptions to enhance soft prompt learning, reducing the distribution gap between plain text and image-text corpus. The authors also propose a training-free approach to mitigate the distribution gap between plain text and image-text corpus by leveraging text descriptions generated from LLMs. Experiments across various settings validate the effectiveness of this approach.

**Strengths:**

**Originality:**

Although the use of weighted query expansions (descriptions) generated by LLMs are not new for retrieval tasks, the authors show its effectiveness for the CLIP model, which might be considered novel in this setting.

**Quality:**

The paper includes experiments on both training-free and prompt tuning. The experimental results in the consistent improvements over the baselines in both settings.

**Clarity:**

The paper is not difficult to follow, but the writing can be clearer.

**Significance:**

The paper proposes both prompt tuning and training-free approaches, which can be useful in practice. The method is also simple to understand and seem relatively easy to implement.

**Weaknesses:**

**Novelty**

The training-free approach is very similar to CBD. The novelty of this is from the learnable weights for the descriptions, which is not training-free. I feel that the training-free strategy is a contribution of CBD. The author should explain why their method shows improvements in the training-free setting.

**Latency and resources**

To make inference, the method needs to perform multiple forward-passes to obtain the embeddings of the descriptions. This can significantly increase latency of the model. If done in parallel, this still requires more resources, which can be challenging when deploying at scale.

**Writing quality:**

The writing of this paper could be improved. For example, the prompt provided to the LLM for the Q part is not a question, and I am not sure conclude is the right word here.

nit: It would be easier to read if the authors move Table 1 and Table 2 closer to their respective sections.

**Missing prompt selection experiment:**

It would be nice to understand the effect of different prompts on the final evaluation metrics.

**Missing references:**

There are several prior works that try to align the distributions of text and image embeddings. Since the authors also propose prompt tuning approaches, it is worth discussing them to provide references to the readers.

Chen, G., Yao, W., Song, X., Li, X., Rao, Y., & Zhang, K. (2022). Prompt learning with optimal transport for vision-language models. arXiv preprint arXiv:2210.01253. - (ICLR 2023)

Tanwisuth, K., Zhang, S., Zheng, H., He, P., & Zhou, M. (2023). POUF: Prompt-oriented unsupervised fine-tuning for large pre-trained models. arXiv preprint arXiv:2305.00350. - (ICML 2023)

Wang, D., Li, M., Liu, X., Xu, M., Chen, B., & Zhang, H. (2023). Tuning Multi-mode Token-level Prompt Alignment across Modalities. arXiv preprint arXiv:2309.13847. - (NeurIPS 2023)

**Questions:**

1. What are some common examples where introducing descriptions lead to misclassified decisions that are not made by the zero-shot CLIP model?

2. Could you incorporate some uncertainty mechanism of the LLMs into the weighting function?

3. The training-free approach is very similar to CBD. CBD uses a mixture of the scores whereas your method uses a mixture of descriptions. Could you explain more on why your method shows improvement?

4.  Can the authors explain why the aggregation is performed on the descriptions themselves instead of the scores? According to CBD, aggregating over scores offers more interpretability.

The paragraph below is taken from CBD.

"We note that one difference from ProtoNets (Snell et al., 2017) is that we compute the class
score s as the mean similarity to support vectors (i.e., descriptor) rather than the similarity to the
mean of the support vectors. Aggregating similarities enables interpretability, as the class score can
be decomposed into similarity with each descriptor, more akin to Chen et al. (2019); Vinyals et al.
(2017). Comparing to the mean of support vectors is essentially the approach of typical “prompt
ensembling.”

5. Is it possible to construct Figure 1 from real data points? It would be more convincing.

---

### Official Review · Reviewer_W7r5 · 2023-10-22

**Soundness:** 2 fair
**Presentation:** 2 fair
**Contribution:** 2 fair
**Rating:** 3
**Confidence:** 5

**Summary:**

This paper focuses on enhancing the image classification performance of vision-language models like CLIP. It introduces an innovative training-free strategy to reduce the distribution gap between plain text and image-text data, leveraging pretrained models such as GPT-3. The paper also presents a novel few-shot learning pipeline that incorporates prompt learning and reweighting to dynamically mitigate distribution gaps between images and text. These contributions aim to address the limitations of existing prompting techniques by providing a more effective and interpretable solution for image classification tasks.

**Strengths:**

1. The method proposed in this paper is straightforward, and the authors present the method clearly.
2. The interpretability of this method is relatively high.

**Weaknesses:**

Motivation: The logical chain of motivation in this paper is incomplete, as detailed below.
1. The last part of the first paragraph of the Introduction lacks coherent logic. Is there any evidence, such as research papers or experimental results, to support the claim that poor interpretability of soft prompting leads to "undermines the reliability of predictions and impedes improvements of the results"?
2.	Is there any literature or experimental evidence to support Figure 1?
3.	If there indeed exists a significant gap between plain text and image-text corpus, then a cross-modal soft prompting approach like MaPLe appears to be a more reasonable choice. It has the capability to simultaneously adjust the model from both modalities, dynamically reducing the distribution gap between text and image modalities. The authors need to explain how the proposed method compares to MaPLe, highlighting the disadvantages of MaPLe (beyond poor interpretability) and the advantages of the proposed approach.

Novelty: In my view, the method proposed in this paper primarily combines the use of text prompts generated by large models with soft prompting. This essentially replaces manually designed text prompts with prompts generated by large pre-trained language models. The utilization of soft prompting is also a common practice. These two aspects, in my opinion, do not constitute fundamental innovation.

Experiment: The experimental section of the paper has significant flaws, as outlined below:
1.	Related works on Prompt Learning based on CLIP, such as CoOp, CoCoOp, and MaPLe, conducted experiments on 11 datasets. Why does this paper only experiment on 7 of them?
2.	The paper lacks an extensive comparison with several recent relevant methods. For example, in a training-free setting the paper should also compare with the following papers:
[1] Ge Y, Ren J, Gallagher A, et al. Improving Zero-shot Generalization and Robustness of Multi-modal Models[C]//Proceedings of the IEEE/CVF Conference on Computer Vision and Pattern Recognition. 2023: 11093-11101.
[2] Udandarao V, Gupta A, Albanie S. Sus-x: Training-free name-only transfer of vision-language models[C]//Proceedings of the IEEE/CVF International Conference on Computer Vision. 2023: 2725-2736.
For example, in a few-shot setting the paper should also compare with the following papers. Among these methods, CoCoOp and MaPLe are well-known, and I believe that every researcher working on CLIP-based Prompt Learning has read these works. The authors even referenced CoCoOp in the paper, so I really can't understand why they didn't compare with them.
[1] Zhou K, Yang J, Loy C C, et al. Conditional prompt learning for vision-language models[C]//Proceedings of the IEEE/CVF Conference on Computer Vision and Pattern Recognition. 2022: 16816-16825.
[2] Khattak M U, Rasheed H, Maaz M, et al. Maple: Multi-modal prompt learning[C]//Proceedings of the IEEE/CVF Conference on Computer Vision and Pattern Recognition. 2023: 19113-19122.
[3] Chen A, Yao Y, Chen P Y, et al. Understanding and improving visual prompting: A label-mapping perspective[C]//Proceedings of the IEEE/CVF Conference on Computer Vision and Pattern Recognition. 2023: 19133-19143.
[4] Wang Z, Liang J, He R, et al. Improving Zero-Shot Generalization for CLIP with Synthesized Prompts[C]//Proceedings of the IEEE/CVF International Conference on Computer Vision. 2023: 3032-3042.
3.	Many crucial ablation experiments are missing in the paper. Here are a few examples:
a)	Experiments using larger pre-trained CLIP models as the backbone are absent. This is important because larger pre-trained models exhibit stronger text-image alignment capabilities. It's worth questioning whether the proposed method remains effective when using larger models.
b)	It would be interesting to see the results of experiments where LLM-generated prompts are used for initialization, followed by soft prompting. Could this approach lead to further improvements?
c)	The paper doesn't provide ablation experiments for several essential hyperparameters, such as the number of text prompts generated for each category (n) and the count of soft prompts (p and q). Understanding the impact of these hyperparameters is vital.
4.	The experimental setup and result calculations are not adequately described. Typically, zero-shot or few-shot methods based on CLIP would partition the dataset into "base" and "novel" classes to separately evaluate zero-shot and few-shot performance. In this paper, there is only one result presented, and there's a lack of accompanying explanations, which is highly confusing.

Presentation: The paper provides a clear description of the method, but it contains multiple typos:
1.	The last sentence in the first paragraph of section 3.2.: “Figure ??”
2.	In the last paragraph of section 3.3: m, n.
3.	In Implementation Details of section 4.1: m, n should be p, q.
4.	In Table1, Table2, Section 4.4, “FVGC” should be “FGVC”.

**Questions:**

Refer to the "Weaknesses" section above.

---

### Official Review · Reviewer_sZnc · 2023-10-22

**Soundness:** 2 fair
**Presentation:** 2 fair
**Contribution:** 1 poor
**Rating:** 3
**Confidence:** 5

**Summary:**

The paper has addressed the issue of suboptimal performance caused by discrete prompts in prompt tuning, as well as the problem of interpretability being absent in soft prompts. It proposes the utilization of LLM (Large Language Model such as ChatGPT or GPT3) to generate text descriptions instead of class names only that align more closely with the distribution of images. In the context of few-shot learning, there is an opportunity to enhance the alignment with the distribution through the adoption of a reweighting strategy for text prompts. The proposed approach demonstrates commendable performance in terms of interpretability, zero-shot learning, and few-shot image classification tasks.

**Strengths:**

- The motivation behind combining soft and discrete prompts is both reasonable and intuitive. It aims to achieve higher interpretability and improved performance. By integrating these two approaches, there is a promising opportunity to enhance the interpretability of the system while also benefiting from the effectiveness of discrete prompts.
- Leveraging the inherent prior knowledge in LLM to aid in image classification tasks is indeed intriguing. It presents an interesting avenue to utilize the pre-existing knowledge within the language model to enhance the accuracy and performance of image classification.

**Weaknesses:**

- From a methodological perspective, the methods mentioned in this paper have been previously explored in other research papers. For example,
    - leverage GPT-3 to produce textual inputs for prompting CLIP with rich downstream linguistic semantics (CVPR 2023: Prompt, Generate, Then Cache: Cascade of Foundation Models Makes Strong Few-Shot Learners ) or
    - prompt a large language model (LLM) to generate informative language descriptions for object classes (ICML 2023 Multi-Modal Classifiers for Open-Vocabulary Object Detection).

    Hence, the idea of utilizing LLM to generate text descriptions that better align with image content is not necessarily a novel concept.

- From the perspective of experimental results, there are a few notable limitations in the paper.
    - Firstly, the paper lacks performance results for four commonly used image classification datasets (UCF101, SUN397, Caltech101, and StanfordCars) from the CoOp dataset. It would be helpful to know how the proposed method performs on these specific datasets.
    - Secondly, the paper only provides results for few-shot/zero-shot experiments, but it lacks generalization experiments such as base-to-new and variants of ImageNet. These types of experiments are also important in evaluating the overall performance and robustness of the proposed method.
    - Lastly, the paper falls short in terms of comparing the proposed method with the latest advancements in the field. While CoOp and CLIP are considered as basic baselines for few-shot/zero-shot settings, there have been subsequent improvements in various research papers. The absence of discussions and comparisons with these advancements weakens the overall claims made by the authors.
- From an interpretability standpoint, one limitation is that the paper does not provide the corresponding descriptions for the trained soft prompts. Without this information, it becomes difficult to understand the specific nature and characteristics of the soft prompts generated during training. Consequently, the overall interpretability of the approach remains limited when considering the entire prompt framework.

**Questions:**

See weaknesses.